# Tachycardia in hyperthyroidism: Not so common

**Muriel Tania Go, Amrutha Mary George, Bettina Tahsin, Leon Fogelfeld**[ID]*

John H. Stroger Jr., Hospital of Cook County, Chicago, Illinois, United States of America

* lfogelfeld@cookcountyhhs.org

## Abstract

### Objective

The commonly held association of hyperthyroidism with sinus tachycardia and widened pulse pressure (PP) has not been reassessed in decades despite patients with hyperthyroidism in current practice not always present with these signs. The study objective was to assess prevalence and variability of sinus tachycardia and widened PP in present day among individuals with different degrees of hyperthyroidism.

### Methods

Data was collected retrospectively from 248 adult patients in an outpatient setting with biochemical evidence of hyperthyroidism, recorded heart rate (HR) and blood pressure (BP) who were not treated with medications that can influence these parameters.

### Results

Mean age was 42.0 ± 14.2 years with 66.9% being female. Median free thyroxine (fT4) level was 3.49 (IQR 2.42–4.58) ng/dL and thyroid stimulating hormone (TSH) 0.02 (IQR 0.01–0.03) mIU/L. Tachycardia, defined as HR >100 bpm, was present in 28.2%. In the lowest and highest fT4 quartiles, tachycardia was present in 16.4% and 38.7% respectively. Using logistic regression, tachycardia was associated with higher fT4 and diastolic BP. More lenient outcome of tachycardia with HR >90 bpm was seen in 47.2%. Widened PP, defined as >50 mmHg, was observed in 64.1% of patients and correlated with higher fT4 and BP.

### Conclusions

Tachycardia is not a common feature of hyperthyroidism today. The relatively infrequent finding of tachycardia in this study compared to older studies may reflect differences in the way medicine is practiced today. The increased ordering of thyroid function tests most likely unmasked cases of mild or asymptomatic thyrotoxicosis. A widened PP was a more prevalent clinical finding in this study.

**Data Availability Statement:** All relevant data are within the paper and its Supporting information files.

**Funding:** The author(s) received no specific funding for this work.

**Competing interests:** The authors have declared that no competing interests exist.

## Introduction

Thyroid hormones are well known to have a significant impact on cardiac function through their effect on cardiac gene expression and sensitivity of the sympathetic system [1]. Extensive evidence indicates that even minimal but persistent changes in circulating thyroid hormone levels lead to changes in heart rate (HR) and contractility [2–4]. Consequently, excessive or deficient thyroid hormones lead to profound changes in cardiac function regulation and cardiovascular hemodynamics.

It is a well-established fact that hyperthyroidism enhances myocardial contractility resulting in increased stroke volume and reduces systemic vascular resistance [5]. This combined effect leads to a widened pulse pressure (PP) [6]. Sinus tachycardia, defined as a HR of >100 beats per minutes (bpm), is another common sign of hyperthyroidism present in 70–100% of patients [7–9]. However, the literature on this finding is from the 1980s with tachycardia inconsistently defined and, in some studies, reported as ≥ 90 bpm. In those studies, mean HR ranged from 95 to 105 bpm [7–9].

In our clinical practice, we noted the absence of sinus tachycardia in many newly diagnosed patients with untreated hyperthyroidism, even with elevated free thyroxine (fT4) levels more than two- to four-fold the upper limit of normal. This clinical observation together with the paucity of recent literature on this subject prompted us to explore the prevalence and range of variability of sinus tachycardia and PP in patients with hyperthyroidism of varying degrees of severity.

## Methods

### Study design

We undertook a retrospective cross-sectional study of adults who were biochemically hyperthyroid with available HR and blood pressure (BP) measurements. Data were obtained from the electronic medical records of outpatients seen in the Cook County Health System and John H. Stroger Jr Hospital of Cook County in Chicago, Illinois. Trained nursing staff in the various clinics using the same standard automated equipment measured the heart rate and blood pressure values. The study was approved by the hospital's institutional review board committee with a consent waiver.

### Inclusion and exclusion criteria

Inclusion criteria included adults >18 years old seen in the outpatient clinics during a two-year period who were biochemically hyperthyroid, defined as having a fT4 >21.1 pmol/L (reference range 7.9–21.1 pmol/L) and thyroid stimulating hormone (TSH) <0.1 mIU/L (reference range 0.34–5.60 mIU/L), and with available corresponding HR and BP measurements recorded within 30 days of thyroid test dates.

Exclusion criteria included patients treated with any of the following medications listed within 90 days of the thyroid tests: levothyroxine (LT4), beta-blockers, digoxin and other antihypertensive drugs including calcium channel blockers, angiotensin-converting enzyme inhibitors, angiotensin receptor blockers, hydralazine, aldosterone receptor blockers and diuretics. Patients treated with methimazole (MMI) or propylthiouracil (PTU) within 30 days of the thyroid tests were also excluded from the study.

### Study group

We obtained data from 141,439 observations of 59,400 patients who had thyroid function testing done during a two-year period (Fig 1). The study population was narrowed to 35,681

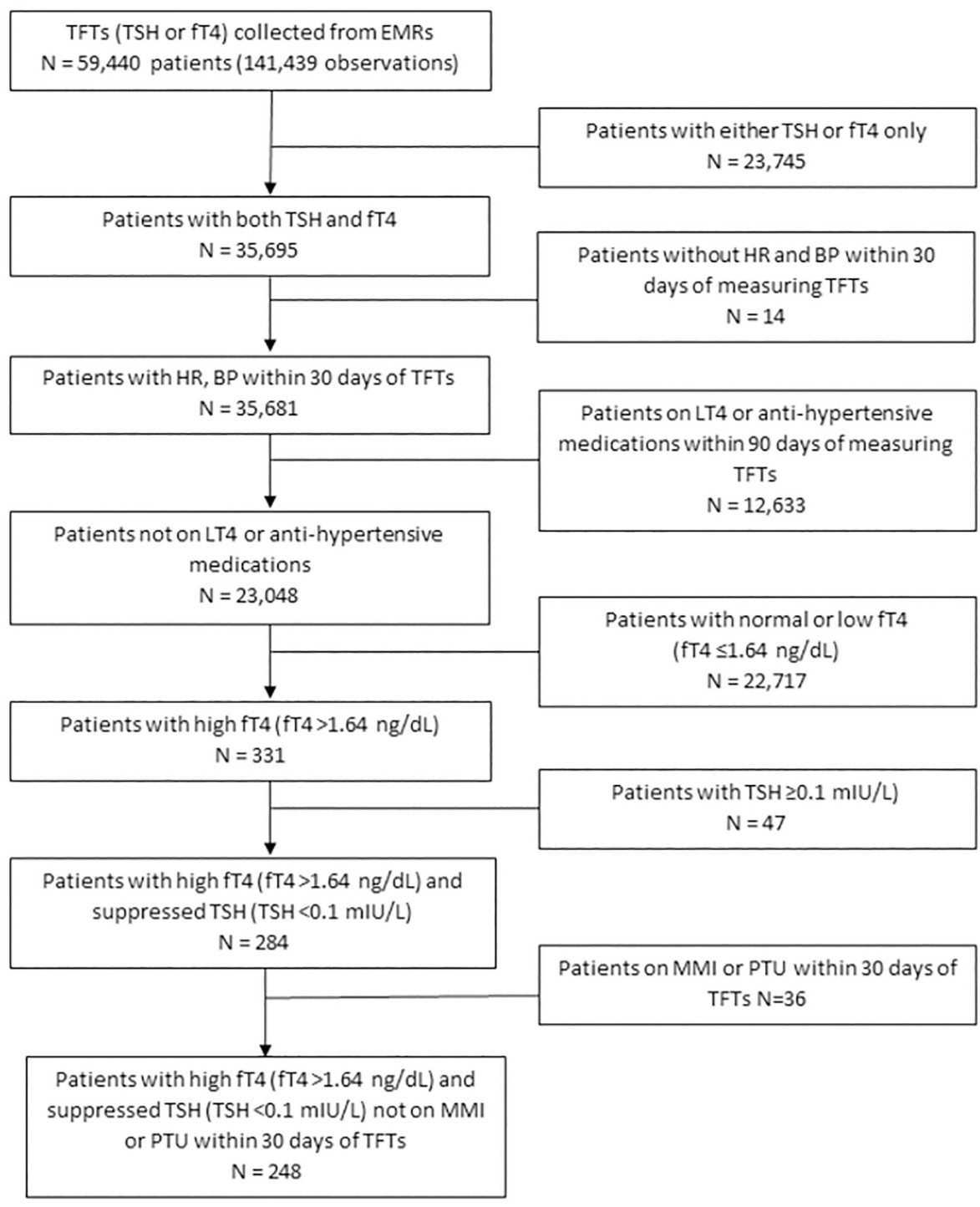

**Fig 1. Flow chart for inclusion and exclusion of the study population with hyperthyroidism.**

patients who had both TSH and fT4 levels and HR and BP measurements within 30 days of measuring the thyroid function tests (TFTs). Fig 1 further explains the exclusion steps based on LT4, beta-blockers, digoxin, and anti-hypertensive medications within 90 days of TFTs. Patients with normal or low fT4 (fT4 ≤1.64 ng/dL) and/or TSH ≥0.1 mIU/mL were also

excluded. The last exclusion factor was the use of anti-thyroid drugs within 30 days of TFTs. The final sample population was 248 patients. For each patient, a detailed chart review was performed, including confirmation of the diagnosis of hyperthyroidism based on laboratory values, functional imaging (if performed), and clinic notes. For patients with two or more documented HR and BP measurements during the clinic visit, the values were averaged.

## Outcomes

The main outcomes related to the levels of thyroid hormones were HR, BP, and PP. For HR, sinus tachycardia was defined as HR >100 bpm [10]. Previous literature has inconsistently defined tachycardia with mean HR ranging between 90–105 bpm [8–11]. Taking into account an inherent error of 5% in measurements, we calculated a second more lenient outcome of tachycardia with HR >90 bpm.

A widened PP was defined as >50 mmHg. Although there were no cutoff values delineating normal levels for different age groups, a widened PP of more than 50 mmHg has been suggested [12]. For older hypertensive individuals, a PP of >60 mmHg has been linked to increased cardiovascular disease-related mortality [13, 14]. We therefore defined pulse pressure of >60 mmHg as an elevated PP in this study.

## Statistical analysis

Baseline characteristics were presented as means ± SD for normally distributed values or as medians with interquartile ranges (IQR). Categorical data were presented by frequency and percentage. Comparisons between interquartile groups were made using ANOVA. To determine the relationship between variables, bivariate correlation, and multivariate linear and logistic regression analyses were used. Two-tailed P values less than 0.05 were considered statistically significant. Data were analyzed using IBM SPSS statistics, version 24.0.

## Results

The study group included 248 patients, 67.8% of whom were females (Table 1). Mean age was 42.0 ± 14.2 years ranging from 19–89 years of age. Median fT4 level was 3.49 (IQR 2.42–4.58) ng/dL and median TSH was 0.02 (IQR 0.01–0.03) mIU/L. Mean HR was 91.0 ± 19.2 bpm with

**Table 1. Clinical characteristics of the study group.**

|  | Patients (N = 248) | Range |
|---|---|---|
| Age (years) | 42.0 ± 14.2 | 19–89 |
| Female, n (%) | 168 (67.7) | |
| fT4 (ng/dL) | 3.49 (IQR 2.42–4.58) | 1.65–6.00 |
| TSH (mIU/mL) | 0.02 (0.01–0.03) | 0.010–0.09 |
| Heart rate, bpm | 91.0 ± 19.2 | 44–151 |
| Tachycardia, HR >100 bpm, n (%) | 70 (28.2) | |
| HR >90 bpm, n (%) | 117(47.2) | |
| Systolic BP, mm Hg | 126.2 ± 17.5 | 88–180 |
| Diastolic BP, mm Hg | 69.7 ± 12.0 | 15–121 |
| Pulse pressure, mm Hg | 56.5 ± 14.4 | 19–105 |
| Widened PP, >50 mm Hg, n (%) | 159 (64.1) | |
| Elevated PP, >60 mm Hg, n (%) | 86 (34.7) | |

Data presented as mean ± standard deviation (SD), number (percentage), or median (interquartile range).

**Table 2. Thyroid function and cardiac values per fT4 quartile.**

| fT4 Quartile (Range in ng/dL) | 1 (1.65–2.38) | 2 (2.42–3.48) | 3 (3.49–4.57) | 4 (4.58–6.00) |
|---|---|---|---|---|
| Patients, n | 61 | 63 | 62 | 62 |
| Mean HR, bpm | 82.6 ± 15.2 | 89.7 ± 18.2 | 93.3 ± 16.7 | 98.4 ± 22.7 |
| Median HR, bpm | 82.0 (72.0–92.0) | 88.0 (79.0–100.0) | 95.0 (79.0–107.5) | 97.5 (85.5–111.0) |
| Mean PP, mm Hg | 51.7 ± 13.8 | 55.1 ± 11.6 | 55.8 ± 14.4 | 63.2 ± 15.2 |
| Median PP, mm Hg | 51.0 (42.5–59.5) | 54.0 (49.0–60.5) | 57.5 (45.8–66.0) | 62.0 (53.5–73.0) |
| Tachycardia, bpm, n (%) | 10 (16.4) | 15 (23.8) | 21 (33.9) | 24 (38.7) |
| HR >90 bpm, n (%) | 18 (29.5) | 23 (36.5) | 37 (59.7) | 39 (62.9) |
| Widened PP, >50 mm Hg, n (%) | 30 (49.2) | 41 (65.1) | 39 (62.9) | 49 (79.0) |
| Elevated PP, >60 mm Hg, n (%) | 13 (21.3) | 16 (25.4) | 25 (40.3) | 32 (51.6) |

tachycardia defined as HR >100 bpm, present in 28.2%. HR >90 bpm was present in 47.2%. The mean PP was 56.5 ± 14.4 mm Hg with 64.1% having a widened PP of >50 mm Hg.

The thyroid and cardiac values per fT4 quartile are shown in Table 2 with HR distribution across fT4 quartiles shown in Fig 2. In the first two quartiles, tachycardia was present in 16.4% and 23.8%, respectively, compared to 33.9% and 38.7% in the third and fourth quartile. There were statistically significant differences in HR between the quartiles by one-way ANOVA [F (3, 244) = 8.0, p <0.001]. The interquartile variability range of heart rate within each fT4 quartile ranged from 20–28.5 bpm.

In bivariate correlation analysis, fT4 positively correlated with HR, PP and systolic BP, and negatively correlated with age (p <0.005 for each variable). In multivariate regression analysis,

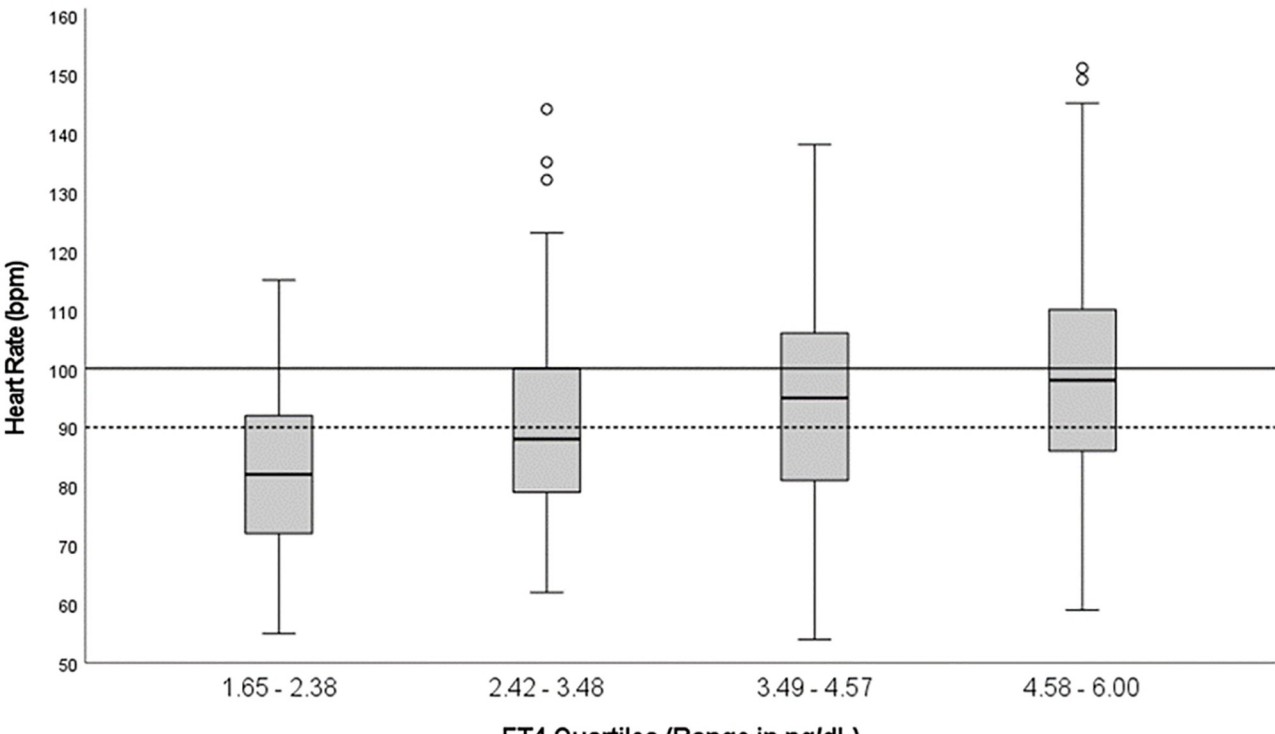

**Fig 2. Median heart rate and IQR in four fT4 quartiles: Solid line represents tachycardia (HR >100 bpm), whereas broken line denotes HR >90 bpm.**

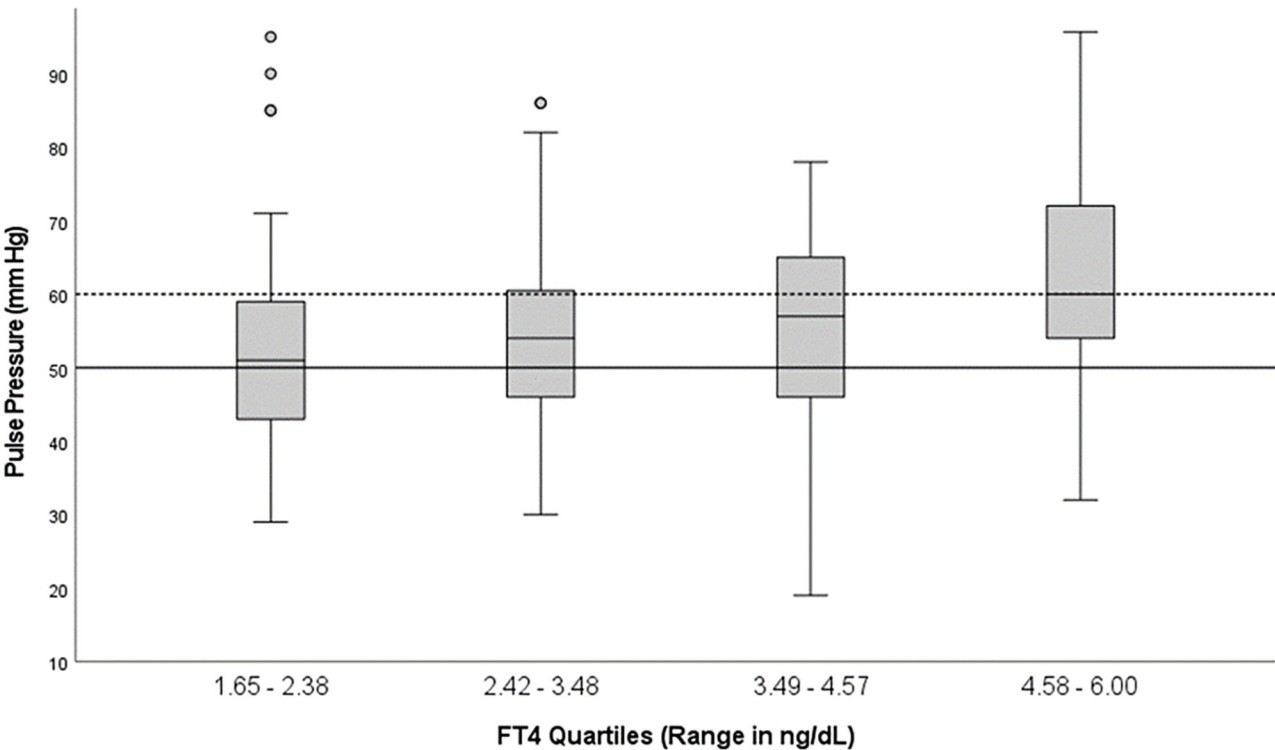

**Fig 3. Median pulse pressure and IQR in four fT4 quartiles: Solid line represents widened pulse pressure >50 mm Hg, whereas broken line denotes elevated pulse pressure >60 mm Hg, a known risk factor for cardiomyopathy.**

increased heart rate correlated with younger age (p = 0.017), higher fT4 (p <0.001), and higher diastolic BP (p <0.001) [F (3, 244) = 17.21, p <0.001, $R^2$ = 0.18]. The presence of sinus tachycardia >100 bpm in logistic regression analysis correlated with higher fT4 (OR 1.03, 95% CI 1.01–1.05) and diastolic BP (OR 1.03, 95% CI 1.01–1.06).

A widened PP (Fig 3) was observed in more than 50% of patients in all quartiles and over 75% in the fourth quartile. In contrast, elevated PP (>60 mm Hg) was observed in fewer than 50% in the first three quartiles and 51.6% in the fourth quartile. There were statistically significant differences in pulse pressure between the quartiles based on one-way ANOVA [F (3, 244) = 7.60, p <0.001].

Widening PP correlated with higher fT4 (p = 0.005), systolic BP (p <0.001) and lower diastolic BP (p = 0.030) using bivariate correlation analysis.

## Discussion

This study evaluated the prevalence and variability of tachycardia in patients with hyperthyroidism. The incidence of tachycardia defined as HR >100 bpm was 28.1% in this study, much lower than the incidence of 70–100% previously reported [7–9, 11, 15]. Given the inconsistent definitions of tachycardia in older studies, we calculated a second scenario of tachycardia with HR >90 bpm and still the incidence was fewer than half of patients.

In the hyperthyroid state, chronotropic alterations manifesting as sinus tachycardia, atrial fibrillation, and shortened PR interval, and inotropic alterations including increased cardiac index, stroke volume and pulse pressure have been well described [16–18]. Early literature with studies conducted between 1943 and 1945 found 72% of thyrotoxic patients had HR of

100–140 bpm [19]. This was comparable to later studies done from 1960–1988 which showed tachycardia in 70–84% of hyperthyroid patients [18]. In a more recent study conducted in a cohort of 3,049 patients [20], while the incidence of tachycardia was not measured, the incidence of other hyperadrenergic signs were 50% lower compared to the 1945 study. Due to a lack of recent literature, we studied the relative prevalence of cardiovascular signs in hyperthyroid patients.

Our study showed tachycardia was present in fewer than one-third of patients. Even if the heart rate threshold were to be lowered for better comparison to previous studies with inconsistent definitions of tachycardia, a little more than half of patients still had HR less than 90 bpm. This significant decline in the prevalence of tachycardia in recent times may be due to the increasing utilization of readily available and highly sensitive thyroid function assays [21, 22]. This widespread change in clinical practice may lead to diagnoses of hyperthyroidism in patients with fewer symptoms. In the earlier studies from 1940 to 1980, there may have been ascertainment bias where only patients who presented with symptoms and advanced thyrotoxicosis were subjected to the then cumbersome thyroid tests available, making tachycardia much more common during those times.

In this study, the level of fT4 was the major determinant of heart rate. However, even in the highest quartile of fT4, tachycardia occurred in only 40.0% of patients. If the HR threshold was lowered to >90 bpm, then the percentage increased to 63.1% in the fourth quartile. The inconsistent effect of hyperthyroidism on HR was also shown in hyperthyroid animal studies. Isolated heart preparations in different study settings noted tachycardia was abolished while the inotropic effect persisted [23–25]. In this study, as in other studies, older age was related to a lower heart rate [20, 26–28].

The hyperthyroid state causes predictable decreases in systemic vascular resistance and diastolic BP and increases in systolic BP and cardiac output [5, 29]. This leads to a widened PP which has been described in thyrotoxic patients in previous literature [30, 31]. In this study, 63.5% of patients and 78.5% in the highest fT4 quartile had widened PP. Based on these findings, one can postulate that a widened PP may be a useful physical finding in the clinical diagnosis of hyperthyroidism. Elevated PP (>60 mm Hg), which has been shown to be associated with heart failure and myocardial infarction [32, 33], was found in 33.8% of our patients and positively correlated with levels of fT4.

In this study, we found wide variabilities in heart rate and pulse pressure within each fT4 quartile, with interquartile ranges varying from 20 to 26 bpm for heart rate and 14 to 20 mm Hg for pulse pressure (Figs 2 and 3). In fact, in clinical practice, it has been found that the intensity of thyrotoxic symptoms and the level of thyroid hormones do not correlate so well [34]. Thyroid function test results may be markedly abnormal in patients exhibiting only mild symptoms, or conversely, mild to moderately abnormal levels can be seen in overtly symptomatic patients [11, 34]. This discrepancy may be secondary to the poor reflection of serum fT4 to the intracellular thyroid hormone concentrations [34], cellular variations in nuclear thyroid hormone receptor sensitivity [35], or differences in adrenergic-mediated hyperthyroid symptoms [36, 37]. However, new data suggests that clinical parameters such as cardiovascular disease had a stronger association of up to 50% with fT4 levels as compared to 23% with TSH levels [38].

The limitation of this study is that it is mainly cross-sectional and does not have longitudinal information of thyroid hormone levels and cardiac parameters, though some patients had more than one observation. Patients who were treated with beta-blockers were excluded and one can assume that beta blockade was used to mitigate the tachycardia and thus many patients with tachycardia were not captured in this study. However, the study was done in a comprehensive public health clinical setting with big central lab, and it is conceivable that

most patients with hyperthyroidism and tachycardia were captured before the beta-blockers were started. Atrial fibrillation was not found in the study group. One explanation is that some patients are admitted to the hospital because of symptomatic atrial fibrillation and undiagnosed hyperthyroidism and are started with beta-blockade before the thyroid function tests are back and, therefore, excluded from the study. However, the purpose of this study was to evaluate the sinus tachycardia as a response to elevated thyroid hormones and not to include cardiac arrhythmias. Another limitation was the exclusion of total or free triiodothyronine (T3) levels which were unavailable for most patients. Since a minor percentage (11%) of patients with untreated thyrotoxicosis have T3 thyrotoxicosis [39], the absence of T3 testing only minimally impacts the overall result of the study.

The novelty of this study is that literature concerning the prevalence and range of variability of sinus tachycardia and PP has not been re-examined in decades. This study shows clear change in the disease manifestation most likely brought about by the current practice patterns with ease of thyroid testing and its increased usage thus allowing diagnosis of mild or asymptomatic forms of hyperthyroidism. This study shows that tachycardia is not such a common feature of hyperthyroidism today. A widened pulse pressure was a more prevalent clinical finding in this study.

## Supporting information

**S1 Data. Hyperthyroid tachycardia data de-identified.**
(XLSX)

## Author Contributions

**Conceptualization:** Muriel Tania Go, Amrutha Mary George, Leon Fogelfeld.

**Data curation:** Muriel Tania Go, Amrutha Mary George, Bettina Tahsin, Leon Fogelfeld.

**Formal analysis:** Muriel Tania Go, Amrutha Mary George, Bettina Tahsin, Leon Fogelfeld.

**Investigation:** Muriel Tania Go, Amrutha Mary George, Leon Fogelfeld.

**Methodology:** Muriel Tania Go, Amrutha Mary George, Leon Fogelfeld.

**Project administration:** Leon Fogelfeld.

**Resources:** Leon Fogelfeld.

**Supervision:** Leon Fogelfeld.

**Validation:** Muriel Tania Go, Amrutha Mary George, Bettina Tahsin, Leon Fogelfeld.

**Visualization:** Muriel Tania Go, Amrutha Mary George, Bettina Tahsin, Leon Fogelfeld.

**Writing – original draft:** Muriel Tania Go, Amrutha Mary George, Bettina Tahsin, Leon Fogelfeld.

**Writing – review & editing:** Muriel Tania Go, Amrutha Mary George, Bettina Tahsin, Leon Fogelfeld.

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
