## [Decision Letter · Decision Letter 0]

2 Aug 2022

PONE-D-22-07597Tachycardia in Hyperthyroidism: Not So CommonPLOS ONE

Dear Dr. Leon Fogelfeld,

Thank you for submitting your manuscript to PLOS ONE. After careful consideration, we feel that it has merit but does not fully meet PLOS ONE’s publication criteria as it currently stands. Therefore, we invite you to submit a revised version of the manuscript that addresses the points raised during the review process.

Indicate the novelty of the manuscript clearly.Tables should be in accordance with author guidelines.Update the references in accordance with the format and include the latest references. 

We look forward to receiving your revised manuscript.

Kind regards,

Dr. Anu Sayal, Ph.D.

Academic Editor

PLOS ONE

https://journals.plos.org/plosone/s/file?id=ba62/PLOSOne_formatting_sample_title_authors_affiliations.pdf".

Reviewers' comments:

Reviewer's Responses to Questions

**Comments to the Author**

1. Is the manuscript technically sound, and do the data support the conclusions?

Reviewer #1: Yes

Reviewer #2: Yes

2. Has the statistical analysis been performed appropriately and rigorously? 

Reviewer #1: Yes

Reviewer #2: I Don't Know

3. Have the authors made all data underlying the findings in their manuscript fully available?

Reviewer #1: Yes

Reviewer #2: Yes

4. Is the manuscript presented in an intelligible fashion and written in standard English?

Reviewer #1: Yes

Reviewer #2: Yes

5. Review Comments to the Author

Reviewer #1: Please mention the full form of TSH and FT4 in the abstract for the first time.

Please separate methods from findings in the abstract.

Please address the novelty of your study.h

Please mention that you reported categorical variables by frequency and percent.

Please revise the tables and justify them based on the authors' guidelines.

Please update your references. You have references from 1946.

You have to revise some of your references based on the Vancouver style.

With regards,

Reviewer #2: An interesting report in issuing current practice that may effect the previous usual findings. It will be good to have more extensive parameters being reported in this report. The usage of the abbreviation can be reduced to enhanced easy to understand reading among readers.

6. PLOS authors have the option to publish the peer review history of their article (what does this mean?). If published, this will include your full peer review and any attached files.

Reviewer #1: **Yes: **Afagh Hassanzadeh Rad

Reviewer #2: No

---

## [Author Response · Author response to Decision Letter 0]

11 Aug 2022

Please see our rebuttal in italics for every reviewer point indicating also the exact locations of the revisions.

PONE-D-22-07597

Tachycardia in Hyperthyroidism: Not So Common

PLOS ONE

Dear Dr. Leon Fogelfeld,

Thank you for submitting your manuscript to PLOS ONE. After careful consideration, we feel that it has merit but does not fully meet PLOS ONE’s publication criteria as it currently stands. Therefore, we invite you to submit a revised version of the manuscript that addresses the points raised during the review process.

• Indicate the novelty of the manuscript clearly.

• The novelty of the manuscript is added in the abstract objective and in the conclusion (last paragraph).

• Tables should be in accordance with author guidelines.

• Was done.

• Update the references in accordance with the format and include the latest references.

• Was done.

• Was done

• Was done

• Was done

If applicable, we recommend that you deposit your laboratory protocols in protocols.io to enhance the reproducibility of your results. Protocols.io assigns your protocol its own identifier (DOI) so that it can be cited independently in the future. For instructions see: https://journals.plos.org/plosone/s/submission-guidelines#loc-laboratory-protocols. Additionally, PLOS ONE offers an option for publishing peer-reviewed Lab Protocol articles, which describe protocols hosted on protocols.io. Read more information on sharing protocols at https://plos.org/protocols?utm_medium=editorial-email HYPERLINK "https://plos.org/protocols?utm_medium=editorial-email&utm_source=authorletters&utm_campaign=protocols"& HYPERLINK "https://plos.org/protocols?utm_medium=editorial-email&utm_source=authorletters&utm_campaign=protocols"utm_source=authorletters HYPERLINK "https://plos.org/protocols?utm_medium=editorial-email&utm_source=authorletters&utm_campaign=protocols"& HYPERLINK "https://plos.org/protocols?utm_medium=editorial-email&utm_source=authorletters&utm_campaign=protocols"utm_campaign=protocols.

We look forward to receiving your revised manuscript.

Kind regards,

Dr. Anu Sayal, Ph.D.

Academic Editor

PLOS ONE

https://journals.plos.org/plosone/s/file?id=ba62/PLOSOne_formatting_sample_title_authors_affiliations.pdf"

Manuscript reviewed and appears to meet the guidelines.

More information about IRB approval and consent waiver provided on page 4 at the end of the Study Design paragraph.

De-identified data file will be included in supplemental materials.

Reference list reviewed and revised in format where need to match Vancouver style. No references were deleted or added.

 

Reviewers' comments:

Reviewer's Responses to Questions

Comments to the Author

1. Is the manuscript technically sound, and do the data support the conclusions?

Reviewer #1: Yes

Reviewer #2: Yes

2. Has the statistical analysis been performed appropriately and rigorously?

Reviewer #1: Yes

Reviewer #2: I Don't Know

3. Have the authors made all data underlying the findings in their manuscript fully available?

Reviewer #1: Yes

Reviewer #2: Yes

4. Is the manuscript presented in an intelligible fashion and written in standard English?

Reviewer #1: Yes

Reviewer #2: Yes

5. Review Comments to the Author

Reviewer #1: Please mention the full form of TSH and FT4 in the abstract for the first time.

Corrected.

Please separate methods from findings in the abstract.

Corrected.

Please address the novelty of your study.

The novelty has been added to the abstract objective and to the end of the discussion in the last conclusion paragraph.

Please mention that you reported categorical variables by frequency and percent.

Added to Statistical Analysis, page 7.

Please revise the tables and justify them based on the authors' guidelines.

On review, the tables appear to meet the table formatting guidelines. Please advise if anything requires revision.

Please update your references. You have references from 1946.

The reference from 1946 is appropriate in the context (Discussion, page 10) of the discussion of past historical findings concerning tachycardia in those with hyperthyroidism.

You have to revise some of your references based on the Vancouver style.

References have been revised.

With regards,

Reviewer #2: An interesting report in issuing current practice that may effect the previous usual findings. It will be good to have more extensive parameters being reports in this report. 

The study reported the outcomes for all the relevant parameters pertaining to hyperthyroidism and tachycardia available for our study group. 

The usage of the abbreviation can be reduced to enhanced easy to understand reading among readers.

Unsure which abbreviation is lacking clarity. Abbreviations after the tables and figures have been removed since it does not appear to fit the format of PLOS One.

6. PLOS authors have the option to publish the peer review history of their article (what does this mean?). If published, this will include your full peer review and any attached files.

Do you want your identity to be public for this peer review? For information about this choice, including consent withdrawal, please see our Privacy Policy.

Reviewer #1: Yes: Afagh Hassanzadeh Rad

Reviewer #2: No

Figures checked via PACE with PACE figures uploaded.

---

## [Editor Report · Decision Letter 1]

15 Aug 2022

Tachycardia in Hyperthyroidism: Not So Common

PONE-D-22-07597R1

Dear Dr. Leon Fogelfeld,

We’re pleased to inform you that your manuscript has been judged scientifically suitable for publication and will be formally accepted for publication once it meets all outstanding technical requirements.

Kind regards,

Dr. Anu Sayal, Ph.D.

Academic Editor

PLOS ONE

---

## [Editor Report · Acceptance letter]

26 Aug 2022

PONE-D-22-07597R1 

Tachycardia in Hyperthyroidism: Not So Common 

Dear Dr. Fogelfeld:

I'm pleased to inform you that your manuscript has been deemed suitable for publication in PLOS ONE. Congratulations! Your manuscript is now with our production department. 

Kind regards, 

on behalf of

Dr. Anu Sayal 

Academic Editor

PLOS ONE